# Investigation of Program Efficiency Overshoot in 3D Vertical Channel NAND Flash with Randomly Distributed Traps

**DOI:** 10.3390/nano13091451

**Published:** 2023-04-24

**Authors:** Chanyang Park, Jun-Sik Yoon, Kihoon Nam, Hyundong Jang, Minsang Park, Rock-Hyun Baek

**Affiliations:** 1Department of Electrical Engineering, Pohang University of Science and Technology (POSTECH), Pohang 37673, Republic of Korea; chrypark@postech.ac.kr (C.P.); junsikyoon@postech.ac.kr (J.-S.Y.); namgee4970@postech.ac.kr (K.N.); hdjang@postech.ac.kr (H.J.); 2SK hynix Inc., Icheon 17336, Republic of Korea; minsang.park@sk.com

**Keywords:** abnormal program cell, charge trap nitride, incremental step pulse programming, Monte−Carlo simulation, over−programming, overshoot, program efficiency, 3D NAND flash

## Abstract

The incremental step pulse programming slope (ISPP) with random variation was investigated by measuring numerous three−dimensional (3D) NAND flash memory cells with a vertical nanowire channel. We stored multiple bits in a cell with the ISPP scheme and read each cell pulse by pulse. The excessive tunneling from the channel to the storage layer determines the program efficiency overshoot. Then, a broadening of the threshold voltage distribution was observed due to the abnormal program cells. To analyze the randomly varying abnormal program behavior itself, we distinguished between the read variation and over−programming in measurements. Using a 3D Monte−Carlo simulation, which is a probabilistic approach to solve randomness, we clarified the physical origins of over−programming that strongly influence the abnormal program cells in program step voltage, and randomly distributed the trap site in the nitride of a nanoscale 3D NAND string. These causes have concurrent effects, but we divided and analyzed them quantitatively. Our results reveal the origins of the variation and the overshoot in the ISPP, widening the threshold voltage distribution with traps randomly located at the nanoscale. The findings can enhance understanding of random over−programming and help mitigate the most problematic programming obstacles for multiple−bit techniques.

## 1. Introduction

Three−dimensional (3D) vertical channel NAND flash memory structures [1,2,3,4] and n−bit multi−level cell (MLC) operation [5] enable storing 2^n^ (n ≥ 2) states in a single cell. The nanostructures consist of cylindrical strings with nanowire thin film transistors vertically stacked, making it easier to store data and improve integration than when the transistors are placed horizontally. Those technologies have successfully helped manufacturers break through the process difficulties of sub−20−nm planar NAND flash. In the planar structure, the bit cost, which refers to the fabrication cost per bit, increases with the cost growth of lithography and etching processes for implementing fine patterns [4]. In the n−bit MLC NAND flash, an incremental step pulse programming (ISPP) algorithm checks whether target cells exceed the program verify (PV) level at every programming pulse. The ISPP scheme controls the tunneling current into the nano−scaled nitride film as intended and narrows the threshold voltage (*V_th_*) distribution [6]. The ISPP slope is defined as the derivative *V_th_* with respect to the program voltage (*V_pgm_*). The ISPP slope is described simply as the threshold voltage difference (∆*V_th_PLS_*) between successive programming pulses (PLS) divided by the program step voltage (*V_step_*), i.e., ∆*V_th_PLS_*/*V_step_*. Previous studies have attempted to increase the program efficiency or ISPP slope [7,8,9]. However, programming (PGM) uniformity or control over the variation in the ISPP slope has become more significant because the *V_th_* gap margin between program states becomes smaller, or even overlaps, as the number of bits per cell increases. Therefore, the uniformity and controllability for variation in PGM operation are crucial because higher bit densities require a narrower width of *V_th_* distribution (*W_Vth_*). This variation is fatal for the penta−(5−bit and 32 PGM states) and the hexa−level (6−bit and 64 PGM states) cell techniques [10,11], which require a narrower *W_Vth_* (100–200 mV) than that of the quadruple−level cell [12]. Photonic integrated circuits (PICs) based on silicon and nitride were introduced to overcome the integration limitation of electronic integrated circuits (ICs), changing the signal medium from electricity to light [13,14]. Still, both ICs and PICs require research on the behaviors of subatomic particles such as electrons/traps and photons for ultrahigh density.

Although various factors cause abnormal program cells (APCs) in the high−density n−bit MLC, quantitatively identifying each cause is difficult. Thus, memory companies generally depend on defense algorithms, such as Bose–Chaudhuri–Hocquenghem [15], and low−density parity−check codes [16] instead of accurately identifying APCs. A simulation−based study analyzed the random behavior of APCs [17], but a huge amount of hardware data is needed to describe their randomness. A previous study has also investigated the change in the number of traps in nitride. It is well known that variations in the number of trapped electrons lead to an expansion of the *V_th_* distribution [18]. In this study, via 3D Monte−Carlo simulation (MCS), the *V_th_* value for the randomly distributed single trap charges was quantitatively confirmed toward the z−axis and radial directions, even when the same number of electrons were trapped. We reflected the capacitance ratio between the tunneling oxide and blocking oxide, including bisecting nitride films that were attached to each other. We also considered the energy band for the different trap positions. Then, 3D MCS was used to predict the probabilistic tunneling outcomes, the *V_th_* distribution, and the overshoot in the ISPP slope. Moreover, the *V_th_* distributions created by the superposition of the over−programming (O−PGM) and read variation were decomposed and analyzed.

Meanwhile, previous studies analyzed the parasitic effects on the *V_th_* distribution and proposed solutions [19,20,21]. They investigated two−dimensional (2D) planar devices with different cell geometries, bias conditions, and cell characteristics. The solutions increase the reading and programming time for use in 3D structures and complicate the circuitry. Compared to planar floating gate devices, 3D NAND flash memory based on charge trap nitride (CTN) can mitigate the floating gate coupling effect [12,22] and upbuild the integration by stacking pairs of wordline (WL) in the upward direction. However, 3D NAND flash suffers from obstacles, such as an additional coupling component after conversion from 2D to 3D [23], as well as large variations in the electrical properties of the cells. These problems are mainly due to geometric effects [24] that do not exist in 2D structures. The previous studies analyzed the pristine or programmed states of devices. However, we decomposed the noisy factors and analyzed how the program procedure was influenced by the geometric effect in the tapered channel pillar [25]. In other words, we examined cell states both within the tunneling process and after the ISPP. The previous research quantitatively calculated how the random variation, which is caused by the geometric effect and the number of preoccupied charges, occurs during the tunneling process [25]. In this work, the extraction methodology was used and further developed to physically analyze how the *V_th_* varies depending on the number of traps and the location when the number of traps is controlled based on the measured average ISPP slope. In particular, we randomized the trap position on a nanoscale using 3D MCS and confirmed the APC behavior with numerous measured cells to support the simulated data.

In this study, we measured *V_th_* distributions in wafer (WF) and its packaged chip that are mass−produced NAND flash. Among the several factors widening the *V_th_* distribution during the ISPP, the O−PGM inducing an ISPP slope of >1 was quantitatively separated. Furthermore, causal factors for the overshoot in the program efficiency and ISPP slope were analyzed using calibrated 3D MCS.

## 2. Materials and Methods

### 2.1. Classification of Abnormal Program Cell Components

#### 2.1.1. Causal Factors Broadening the Threshold Voltage Distribution

To identify the causal factors broadening the *V_th_* distribution, we experimented with a chip and WF of a 3D U−shaped stacked memory array transistor (SMArT) structure [1,3], as shown in Figure 1. *V_th_* and ∆*V_th_PLS_* were read for approximately 70,000–80,000 cells in each step of ISPP and a page unit. The nanowire thin film transistor cell of 3D NAND flash memory comprises metal−oxide−nitride−oxide−polysilicon. Moreover, it is penetrated by a macaroni oxide filler to mitigate the grain boundary effect, as shown in the inset of Figure 1. The basic memory operations act as a page or block unit, and the 3D NAND array is a stacked structure of multiple nanoscale thin film transistors with multiple materials. The stacked cells organize a string connected to a sourceline (SL) that supplies charges and a bitline (BL) that transmits/receives signals. They operate like a source and drain in a logic device, and the BL current is transferred to a page buffer, and the analog signal of the current, or *V_th_*, is converted into digital data, or bit.

Considering the PV level as the stop line for programming, cells may persist to cross the level and span the larger *V_th_* shift than *V_step_*, compared to the ideally programmed cells with *W_Vth_* = *V_step_*. The *V_th_* distributions stretch on both sides simultaneously due to factors such as under−programming (U−PGM) [26] due to the negative overshoot of ISPP slope or charge loss, background pattern dependency (BPD) [19], cell−to−cell interference (CTCI) [20,23], read variation, and intrinsic O−PGM. We experimented with the target cell after defining the data pattern of the adjacent cells with five PGM/erase cycles, the initialization procedure in Figure 2a. Furthermore, we used the ISPP scheme to store multiple bits in cells, as in Figure 2b. After the initialization condition of erasing the adjacent cells, we experimented with the page unit test. Compared to the planar arrays, 3D NAND flash arrays have a thicker pitch between layers and the nonmetallic storage layer, as shown in Figure 2c. Thus, BPD and CTCI can be ignored in our measurement conditions.

Furthermore, U−PGM is neglected because its probability is less than 0.3% in our measurement conditions. We define the APC as the cell programmed to a higher *V_th_* than *V_step_* from the PV level. For n−bit MLC, in which the number of program states becomes two to the power of n, the APC invades the next or higher program state. When the invasion of APC into the *V_th_* of a higher program state exceeds the error correction threshold in the defense algorithms [15,16], the cells overlapping *V_th_* distributions fail to be read. Because APC is harmful for the *V_th_* distribution that stores multiple bits, identifying the causes of APC is essential. Interestingly, APC can be triggered during both the reading and PGM operations. When the target cells are verified incorrectly during, immediately after, or long after the PGM operation, the cells can be read as larger than the data that are actually stored (read variation), or they can be more programmed (intrinsic O−PGM). We analyzed the read variation in the *V_th_* distribution in the next section. Moreover, Figure 2a shows the experimental procedure in this work, and Figure 2c represents the methodology corresponding to the procedure.

#### 2.1.2. Read Variation Effect on Expansion of Threshold Voltage Distribution

To confirm the read variation effect on the expansion of *V_th_* distribution, we ran the chip for at least seven days at room temperature, as in Figure 2a. Thus, retention error caused by material properties in nitride and oxide thin films while reading, including short−term retention within a few seconds [27,28], no longer occurred noticeably (Figure 2c). Then, the widths (*W_RD_*s) of the read variation distribution (∆*V_th_RD_*) were obtained from the difference between the left and right tail bits of the read variation distribution at a 0.5% probability with repeatedly reading one hundred times, similar to previous research [25]. The *W_RD_*s in the chip and WF are represented in Figure 3a.

The *W_RD_* of the chip is larger than that of the WF because the chip was mounted on a board with an additional peripheral circuit, which aided the experiments using numerous cells. However, the *W_RD_* of the chip and WF were fixed regardless of the different *V_step_*s and locations, as in the previous study [25].

Random telegraph noise (RTN) can be observed in the discrete fluctuations between upper and lower boundaries in *V_th_* and channel current at tens of nanometer nodes [29,30,31,32], similar to the structures in Figure 3b,d [33,34]. However, the read variation measured from the entire page is generated by undesirable effects, such as RTN, carrier transport through random grain boundaries in the polycrystalline material [35,36], common source line noise [37], and noise from the peripheral circuitry to support chip measurements [25]. In a previous study [38], the causes of the *V_th_* distribution widening were classified into program error and RTN. RTN was responsible for U−PGM, and the study [39] confirmed that it was programmed below the PV level. However, we considered that RTN also affects regions programmed above *V_step_* from the PV level because the variation of *V_th_* in the read operation occurred even after the program was completed. In particular, reflecting the read variation by the RTN in the previous studies [18,29,30,31,32,33,34,39], and despite the different optimal measurement conditions of RTN [31], we verified that the measured *W_RD_*s of the chip and WF in Figure 3a,c,d are larger than the *W_RD_* of RTN in Figure 3a,b. Hence, we confirmed that non−RTN factors influenced the read operation in Figure 3c,d.

### 2.2. Characterization of Over−Programming in Abnormal Program Cell

After eliminating the read variation from the *V_th_* distribution as in the experimental procedure in Figure 2a, we can determine how the PGM operation itself affects APC [25]. In Figure 4, n indicates the ordinal number for the respective *V_step_*s. We show experiments of ∆*V_th_PLS_* from the (n − 1)th to nth PGM pulse to analyze the difference in *V_th_*. In Figure 4, ∆*V_th_PLS_* is the *V_th_* difference between two consecutive PGM pulses, whereas ∆*V_th_RD_* in Figure 3 and Figure 4 is a variation in *V_th_* during repetitive read operations.

The ∆*V_th_PLS_* distribution of 70,000–80,000 cells was measured based on *V_step_*. We split *V_step_* to observe different trapping occurrences. In other words, a higher *V_step_* will result in more trapping occurrences and a greater probability of O−PGM.

## 3. Results and Discussion

### 3.1. Extraction of Over−Programming and Dependence on the Program Step Voltage

Assume that read variation is the unique cause of APC. Then, ∆*V_th_PLS_* should be distributed as read variation (blue line) centered on the *V_step_* × ISPP slope in Figure 4. However, the real ∆*V_th_PLS_* (black line), was more widely distributed than a read variation (blue line), which indicates that O−PGM in the cell itself is also responsible for APC.

After determining the value of n when the program efficiency begins to saturate, which is when most APCs manifest, the ∆*V_th_PLS_* distributions were obtained and are shown in Figure 4. The ordinal numbers for *V_step_* = 0.75, 1, and 1.25 a.u. are 16, 10, and 8, respectively. In Figure 4a–c, the number of programming pulses decreases as *V_step_* increases because a higher *V_step_* requires fewer pulses to reach a similar PV level. The extraction method of O−PGM from the APC can be described as follows [25], and it produces the subtraction ∆*V_th_RD_* from ∆*V_th_PLS_* on the condition in Figure 2a. The intrinsic O−PGM (∆*V_th_O_*_−*PGM*_) can be obtained through this method.

The extracted O−PGM is expressed as a distribution rather than a single value, so a quantitative index is required to compare how much O−PGM occurred. Therefore, we should define the probability of O−PGM (*f_O_*_−*PGM*_(∆*V_th_*)) and calculate its conditional expected values (*E*[*O*−*PGM*]), which is similar to calculating the conditional expected value in the previous study [25].

We defined the probability functions for ∆*V_th_*s by PGM pulse (*f_PLS_*(∆*V_th_*)), read variation (*f_RD_*(∆*V_th_*)), and O−PGM (*f_O_*_−*PGM*_(∆*V_th_*)). The *f_RD_*(∆*V_th_*) should be translated to the peak of *f_PLS_*(∆*V_th_*), where most cells are distributed on ∆*V_th_PLS_* to extract *f_O_*_−*PGM*_(∆*V_th_*). Therefore, *g_RD_*(∆*V_th_*), the probability of read variation centered on *V_step_*×ISPP slope can be described as the following equation:(1)$$g_{RD}\left(\Delta V_{th}\right)=f_{RD}\left(\Delta V_{th}-V_{step}\times ISPP\;slope\right)$$

The probability of O−PGM can be obtained by the following equation:(2)$$f_{O-PGM}\left(\Delta V_{th}\right)=f_{PLS}\left(\Delta V_{th}\right)-g_{RD}\left(\Delta V_{th}\right),$$
where ∆*V_th_* > *V_step_* and the probability is nonnegative. The extraction procedure is the same as calculating the probability of intersection with the complement of read variation from the total probability of ∆*V_th_PLS_*.

Finally, we can obtain a quantitative value for the *f_O_*_−*PGM*_(∆*V_th_*) that considers the conditional probability so that the summation of the probability is equal to one where ∆*V_th_* > *V_step_*. We used the conditional expected value of *f_O_*_−*PGM*_(∆*V_th_*) (*E*[*O*−*PGM*]) as the following equation:(3)$$E\left[O-PGM\right]=\frac{\int_{V_{step}}^{\infty \;}\Delta V_{th}f_{O-PGM}\left(\Delta V_{th}\right)d\left(\Delta V_{th}\right)}{\int_{V_{step}}^{\infty \;}f_{O-PGM}\left(\Delta V_{th}\right)d\left(\Delta V_{th}\right)},$$
where the APC is in the region of ∆*V_th_* > *V_step_*, as previously defined in Section 2.1.1 and Figure 2a.

Then, we obtained the values of *E*[*O*−*PGM*] = 0.93, 1.12, and 1.35 a.u. for *V_step_* = 0.75, 1, and 1.25 a.u., respectively. Comparing the *E*[*O*−*PGM*], the O−PGM contributes to APC by increasing as *V_step_* increases. The influence of the PGM itself was confirmed by removing the read variation component independent of *V_step_*s. For the intrinsic O−PGM to be ideal, a higher activation energy for detrapping is required than the read variation prone to capture/emission described in Figure 3. It means the O−PGM cells have experienced stronger Fowler–Nordheim tunneling than the process design in ISPP. Furthermore, the measured ∆*V_th_* was affected by the trap characteristics, such as trap locations and energy levels rather than the fixed trap fluctuation that could easily be detrapped/trapped or captured/emitted in the short term. This is analyzed in the next section.

### 3.2. In−Depth Analysis of Over−Programming

The number of traps can be controlled by changing *V_step_* in the measurement. Moreover, we simulated *V_th_* shifts, i.e., O−PGM, in the simulation without the read variation from the repetitive capture/emission and the external noise in Figure 3. MCS is used for the probabilistic approach to phenomena in which the unpredictable and inseparable variables are parametrized [17]. In particular, MCS can implement random and nonuniform trap behaviors due to the variation in nanotechnology manufacturing on the deca−nanometer scale. In this study, 3D MCS obtained the probabilistic distributions determined by the states of traps in the 3D nanowire vertical channel NAND flash structure, such as the number of traps, their locations, and energy levels. They are material properties of nanoscale films that are difficult to observe by measurement. We confirmed the random ISPP slopes in a previous work [25]. Further developed, the program sequence from the beginning to end, as compared to the previous study [25], is shown in Figure 2a and Figure 5a. The measured ISPP slopes (average ~0.8) from the 70,000–80,000 samples in Figure 5a were calibrated using 3D MCS, which randomly changes the number of traps, their locations, and their energy levels, shown in Figure 5b. The accurate number of traps cannot be determined due to randomization, and the change in the trap number causes the overshoot and the undershoot of program efficiency. Thus, we chose the range for shooting and the reference value based on the measured program efficiency. To realize an average ISPP slope of 0.8, 190–265 nitride traps in a cell were occupied by electrons or a single PGM pulse (Figure 5b).

The number of traps was calculated for a nanowire FET device, including fringing capacitances [40]. The fringing capacitances are the extrinsic parasitic capacitances between the gate and other conductive materials beyond the effective channel length, such as the WLs of adjacent cells, the channel regions of the intercell, and the channel below the gate conducted by the fringing field. In other words, the average value of the ISPP slope in 3D MCS was calibrated to match the measurement, as shown in the triangles in Figure 5. APCs had ISPP slopes above 1 and even up to 2. The slopes indicate that the effect of the programming itself was larger than the read variation. In Figure 5, we focused on the cells entering the *V_th_* region from PV + *V_step_* to PV + 2 × *V_step_*, defined as the APC region. Although the dimension of the gate stack and the pitch between the materials are designed at the manufacturing engineering to produce an average ISPP slope, program efficiency overshoot occurs in realistic operation. The Gaussian distribution of the injection probability has been previously defined as the Poisson statistics and *V_th_* distribution by RTN in a floating gate device [18]. We proceeded with the analysis of 3D structures based on CTN, the latest process technology, by extending the analysis performed in previous studies based on the floating gate device [12,18,22,30]. Despite the similar *V_step_* and lower ISPP slope (average ~0.8 compared to the floating gate with 1), the measured *V_th_* distribution is wider than the one described by the Poisson statistics and RTN [18].

There are more APCs for each PGM state of an n−bit MLC in Figure 5a, but only the rightmost APCs of the ∆*V_th_PLS_* distribution are displayed to distinguish them visually due to their many overlaps. The rightmost APCs exhibit the worst case of O−PGM.

Figure 6 displays (a) the energy band diagram of a simulated polysilicon channel and (b) the contour plot showing that ∆*V_th_PLS_* concentrates around the top of the barrier. We assume that material parameters, such as maximum trap concentration, maximum trapping distance from the tunneling oxide for electrons, and the blocking oxide for holes and capture cross−section in the device physics of the simulation, do not change with the trap location. These device parameters affect the Poisson statistics and represent the injection probability for Fowler–Nordheim tunneling [18]. Splitting the parameters confirms the influence of the electron trapping process and the injection probability. However, the objective of the 3D MCS was to confirm the *V_th_* to the positions of the trapped electrons due to the infinitesimal positioning of the traps in the nanoscale film rather than changing the probability of capturing the electrons. Therefore, since we analyzed 3D MCS with electrons trapped, we can analyze the effect of the trapped position of electrons on *V_th_*, as shown in Figure 6b.

For a given number of traps that have randomly distributed trap sites in every PGM pulse, the nitride traps near the top of the barrier of the channel induce a greater *V_th_* shift than traps at other locations. Furthermore, at the nitride/tunneling oxide interface trap, the barrier height based on the energy level of the SL increases significantly. In particular, the barrier height increases the most when the interface trap is near the top of the barrier. Thus, *V_th_* increases, as shown in Figure 6.

Figure 7 shows the energy band diagram of a polysilicon channel enlarged by the channel length of the target cell, and the inset of Figure 7a shows the energy band diagram of a channel string. Trap #2 is located at the same z−coordinate as the top of barrier in the channel. Figure 7b shows the dependence of barrier height on the trapping position from trap sites #1–#5. In particular, trap #2 has the largest barrier height in the channel and can be expected to have the largest *V_th_*, as shown in Figure 6. Positions #1 and #5, far from the top of the barrier, are located approximately at the barrier height of cells without the trap.

Figure 8 shows the increase in *W_PLS_* as the interface trap increases within a 1 Å range from the tunneling oxide to the nitride. Figure 8b shows that *W_PLS_* increases by 0.015–0.031 a.u., and Figure 8c shows that the ratio of the O−PGM to the total ∆*V_th_PLS_* distribution increases by 0.17–1.23% as the ratio of the single interface traps to the total traps in the nitride increases. Datapoints showed that the ratio of the interface traps to the total traps in the absence of interface traps is zero. These data points are omitted in Figure 8b,c because we adopted a semi−logarithmic scale. The traps in the interfacial region and near the intercell migrate more in the lateral direction than in the noninterface region, resulting in a wider left side of the *V_th_* distribution. Because the interface traps are closer to the tunneling oxide than the noninterface traps, *V_th_* is calculated as higher, causing a widening of the right side of the *V_th_* distribution. In addition, the interface traps screen the channel charges from the gate bias and the channel current reduces; then, the *V_th_* becomes larger than bulk nitride traps. Thus, as the number of single interface traps increases, the ∆*V_th_PLS_* distribution widens. Moreover, as shown in the contour spacing of Figure 6b, the difference in ∆*V_th_PLS_* according to the distance from the nitride/tunneling oxide interface is largest at the top of the barrier. The difference decreases as it moves away from the top of the barrier in the z−direction. These results indicate that, although the number of traps is the same, the location of the traps affects the likelihood of a *V_th_* shift occurring. Overall, in the nitride region, the quality of the nitride/tunneling oxide interface is an essential factor in reducing the O−PGM and the extent of the *V_th_* distribution. In particular, more defects at and near the interface (nonideal storage) will manifest in the scale−down process because the mitigation treatment for them is more difficult due to the thermal budget for the multiple materials. To make matters worse, their proportion to total traps, including bulk nitride traps (ideal storage), increases. More O−PGM will therefore take place, and the *V_th_* will broaden, as in Figure 8.

We utilized the method to separate the read variation and O−PGM from the *V_th_* [25]. Predictive 3D modeling of *V_th_* distribution in gate-all-around cylindrical nanowire devices, including parasitic capacitances [40], which requires a large amount of measurement data and quantitatively confirms the influence of each on APCs. As technology nodes evolve and the number of bits stored per cell increases, the *V_th_* gap margin between adjacent program states becomes smaller. Thus, data failures will increase because the overlapped data between the program states exceed the error correction threshold. Furthermore, 3D MCS implements the uncertainty and randomness of traps in 3D nanotechnology manufacturing with shrinking trends. In the next−generation process technology, such as the biconcave nitride [2], designing where to bend can be applied to find the optimal point in minimizing the program efficiency overshoot and reducing process cost. Utilizing the study of trap properties inside and at the boundary between the thin film materials to ensure data reliability will help determine the priority criteria for the PGM and the read strategies to be properly modified.

## 4. Conclusions

The factors that widen the *V_th_* distribution or attribute to the abnormal program cells were quantitatively classified and confirmed by experimental tests from the mass−produced chip and wafer. Furthermore, the intrinsic over−programming from the entire abnormal program behaviors was obtained by the extraction method, then analyzed using 3D MCS based on the electronic properties of the nanoscale thin film, such as the number of traps, their locations, and energy. As traps are closer to the top of the barrier in the conduction band and interface, it is more critical for over−programming despite having the same number of traps as in the nitride volume of the target cell. These material properties of the nanostructure can enhance the understanding of random over−programming and help determine an approach to mitigate the most troublesome program problem in an ISPP scheme for the n−bit MLC technique. We analyzed the impact of the read variation and over−programming on abnormal program cells. Furthermore, the analysis of the failures in verifying operations between program pulses during the ISPP, which is the situation when the programming for cells is successful but redundant program pulses are applied, can be advanced in future research. As a result, the width of the *V_th_* distribution can become smaller and the *V_th_* gap margin wider.

## Figures and Tables

**Figure 1 nanomaterials-13-01451-f001:**
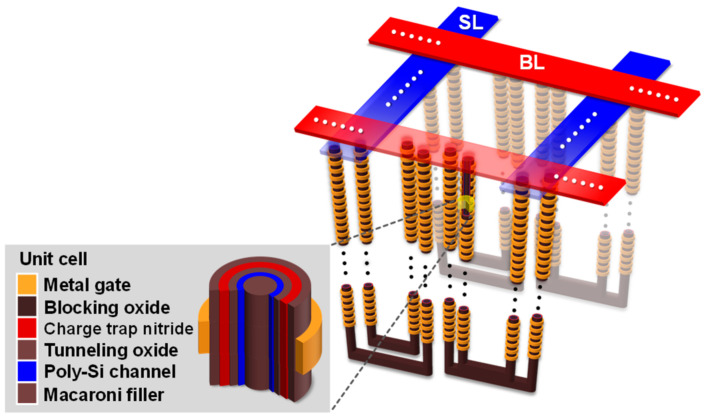
Schematic of a 3D U−shaped SMArT structure (inset: a nanowire thin film transistor cell composed of multiple materials based on charge trap nitride as storage layer). The channel strings are connected to the sourceline (SL) and the bitline (BL), which are the source of electrons, and a signal line connected to the page buffer, which is a circuit that converts the analog channel current into digital data (bits), respectively.

**Figure 2 nanomaterials-13-01451-f002:**
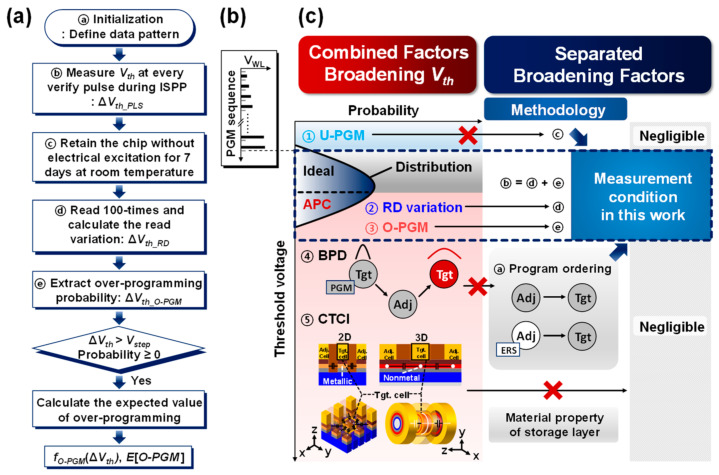
(**a**) The flow chart of the experimental procedure to analyze the program efficiency overshoot in a 3D NAND flash memory chip. (**b**) Wordline voltage (*V_WL_*) versus PGM sequence during ISPP scheme. (**c**) The combined factors broadening *V_th_* (① under−programming ② read variation ③ intrinsic over−programming ④ background pattern dependency from adjacent (Adj.) cells to a target (Tgt.) cell ⑤ cell−to−cell interference) and the separated broadening factors with the controlled measurement condition in this work. The methodology (ⓐ–ⓔ) to exclude ①, ④, and ⑤ corresponds to (**a**).

**Figure 3 nanomaterials-13-01451-f003:**
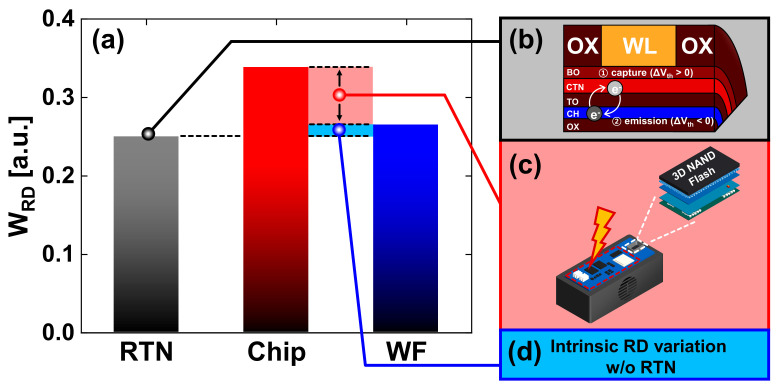
(**a**) Distribution width of read variation in chip and WF obtained by reading iteratively seven days after completion of programming. (**b**) RTN behavior with charge capture and emission. (**c**) Noise from the peripheral circuitry in the chip−level equipment. (**d**) Measured intrinsic read variation in WF except for RTN.

**Figure 4 nanomaterials-13-01451-f004:**
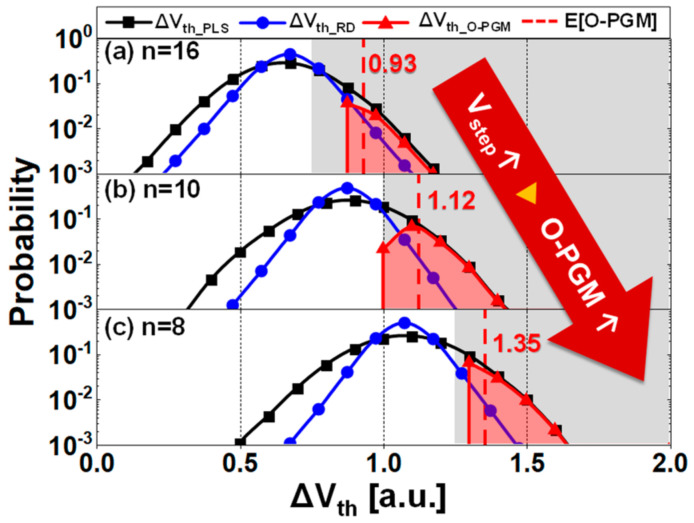
∆*V_th_PLS_* distribution (black lines) from (n − 1)th programming pulse to nth programming pulse during the ISPP, ∆*V_th_RD_* by read variation (blue lines), ∆*V_th_O_*_−*PGM*_ by O−PGM (red lines), and the conditional expected values of O−PGM (red dashed lines) to (**a**) *V_step_* = 0.75 a.u., (**b**) *V_step_* = 1 a.u., and (**c**) *V_step_* = 1.25 a.u., respectively; grey regions represent abnormal program cell regimes.

**Figure 5 nanomaterials-13-01451-f005:**
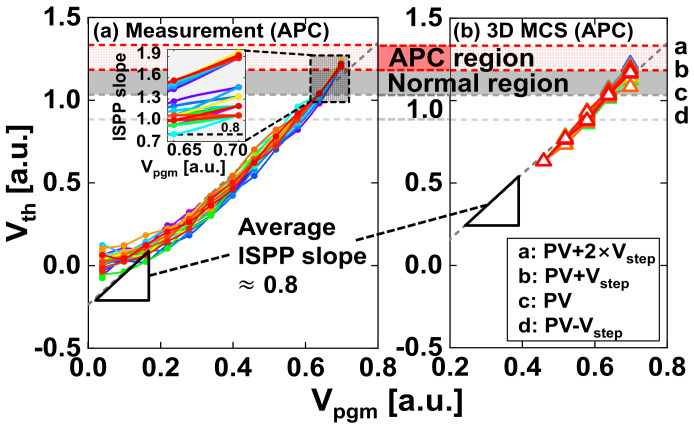
*V_th_* versus *V_pgm_* of APCs in (**a**) measurement and (**b**) 3D MCS. The inset of (**a**) is the overshoot in the ISPP slope for APCs near the PV level. Different colored lines in (**a**,**b**) represent the curves in the rightmost APCs of the ∆*V_th_PLS_* distribution. The measured average ISPP slope was calibrated using 3D MCS considering fringing capacitances. The average ISPP slopes in measurement and 3D MCS are indicated by triangles, respectively. The cells in the *V_th_* region from a (PV + 2 × *V_step_*) to b (PV + *V_step_*) are APCs, the cells from b (PV + *V_step_*) to c (PV) are normal, and the cells from c (PV) to d (PV − *V_step_*) are before passing the programming reference, c (PV).

**Figure 6 nanomaterials-13-01451-f006:**
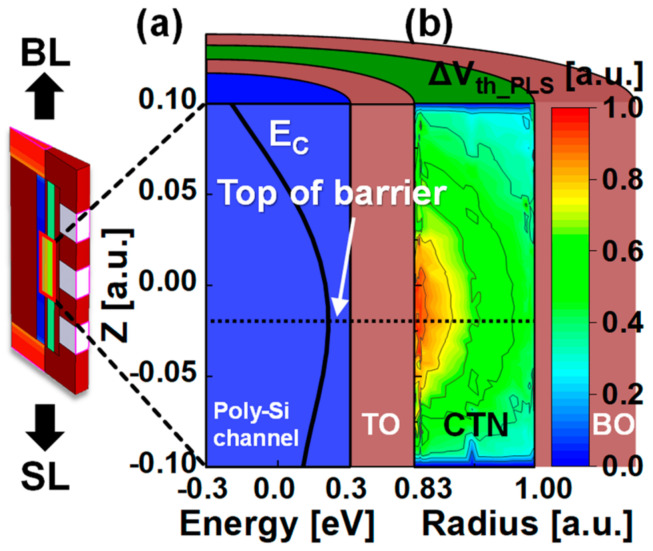
(**a**) Band diagram of the poly−Si channel in the read operation after PGM operation and (**b**) contour plot with the magnitude of ∆*V_th_PLS_* highlighted in different colors at different nitride traps in a target cell in the string.

**Figure 7 nanomaterials-13-01451-f007:**
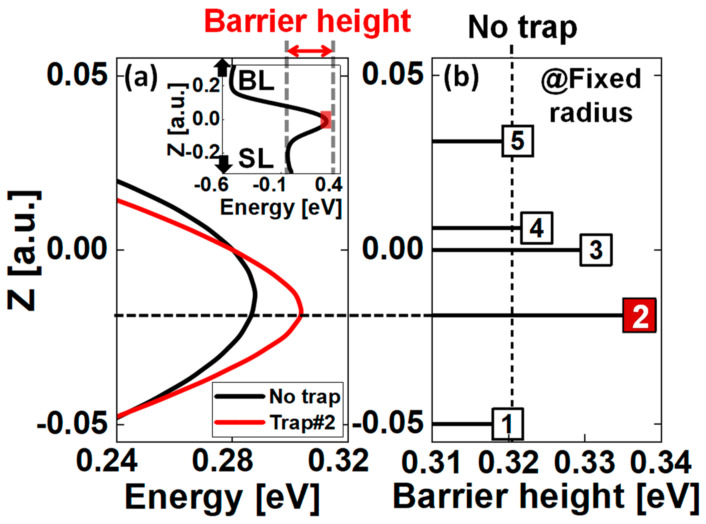
(**a**) Band diagram of a channel enlarged by the channel length of the target cell in an inset, with and without a single trap (or before and after programming) at trap #2 in (**b**), (**b**) change in barrier height with respect to the position of single trap numbered from 1 to 5.

**Figure 8 nanomaterials-13-01451-f008:**
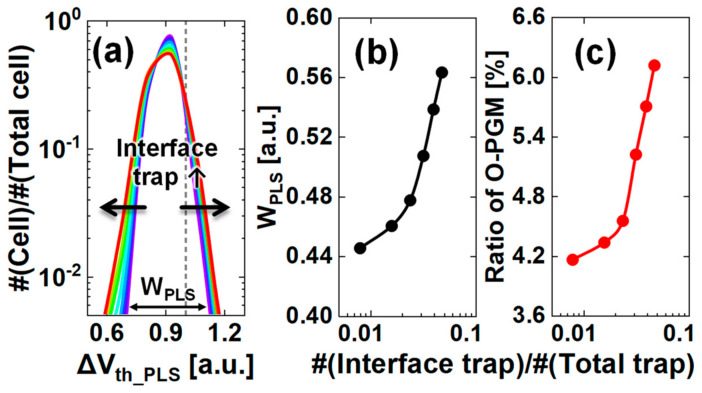
(**a**) Increase in ∆*V_th_PLS_* distribution width with an increasing number of nitride/tunneling oxide interface traps within a 1 Å range. The lines in (**a**) are rainbow−colored from purple to red as the interface trap increases. The distribution broadens in the direction of the arrow as the interface trap increases. The dashed line indicates *V_step_* values obtained experimentally. (**b**) Width of the ∆*V_th_PLS_* distribution (*W_PLS_*) and (**c**) increasing ratio of O−PGM to the number of nitride/tunneling oxide interface traps per total number of traps in the nitride region.

## Data Availability

Data can be made available upon request from the authors.

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
