# Peer review of "Investigation of Program Efficiency Overshoot in 3D Vertical Channel NAND Flash with Randomly Distributed Traps"

_nanomaterials, 2023, doi:10.3390/nano13091451_

Round 1

Reviewer 1 Report

The comments and suggestions for authors are listed in the attached file. 

Author Response

Dear Reviewer 1

Thank you for delicately reviewing our manuscript. We have thoroughly revised the manuscript, considering all referee comments, and we have no disagreement with any comments. Furthermore, we marked up using the “Track Changes” function in MS Word to view any charges easily. We provide our response to all of your comments point-by-point. Please see the attachment.

Best regards,
Chanyang Park et al.

Reviewer 2 Report

In this manuscript (nanomaterials-2320663), author report the program efficiency overshoot in 3D vertical channel NAND flash with randomly distributed traps. By carefully reading this paper, it is easy to discover some good innovations. At the same time, this paper presents sufficient data that can well support the conclusions reached by the author. Therefore, I recommend that this paper can be accepted and published after appropriate revisions.

(1) The experimental section requires some details, and it is recommended that the author should add the device preparation process to the experimental section.

(2) It is a lack of discussion on the working mechanism of the device, and the author should supplement the discussion on the working mechanism of the device.

(3) A small number of grammatical and spelling errors require careful correction by the author during the revision process.

Author Response

Dear Reviewer 2

Thank you for delicately reviewing our manuscript. We have thoroughly revised the manuscript, considering all referee comments, and we have no disagreement with any comments. Furthermore, we marked up using the “Track Changes” function in MS Word to view any charges easily. We provide our response to all of your comments point-by-point. Please see the attachment.

Best regards,
Chanyang Park et al.

Reviewer 3 Report

The paper is interesting and well written.

The analysis performed by simulation is well supported by measurements.

In my opinion, the paper can be published.

Author Response

Dear Reviewer 3

Thank you for delicately reviewing our manuscript. We have thoroughly revised the manuscript, and it was helpful to respond to the comments of other reviewers by highlighting your rating comments. Thank you.

Best regards,
Chanyang Park et al.

Reviewer 4 Report

The paper by Park et al., titled as “Investigation of Program Efficiency Overshoot in 3D Vertical Channel NAND Flash with Randomly Distributed Traps” is devoted to the three-dimensional  Monte-Carlo  simulation which would assist in cognition of  physical  origins  of  over-programming of flash memory cells during programming voltage pulse. The over-programming tendency is considered as a technical issue related to the content of electronic traps in  a real material applied as charge trapping layer in such memory cells, silicon nitride. Additional trap sites are present, besides the relatively narrow band gap characteristic of the charge trapping layer, nearby the band edges and are, assumptionally, randomly distributed in that nitride The structure and order in the 3D memory strings are drawn clearly and the issues raised, in principle, clearly enough.

It is suggested that the authors carefully double check the presentation of the final results. It appears somewhat confusing to read the following sentence in conclusions: ”A  single  trap  near  the  top  of  the  barrier  in  the  conduction  band  and  interface  was  critical  for  over-programming  despite  the  same  number  of  traps.” What is actually meant by such a sentence? Why is “single trap” considered as equivalent to the expression “the same number of traps.” Could the authors elaborate the meaning of the conclusions so that it becomes understandable for a reader?

One might say, that the connection between simulation and real nanomaterials is somewhat loose in this manuscript.  However, the material properties are addressed, nevertheless, and understanding  of  the interfacial behaviour between the constituent materials layers in such nanostructure  may allow one to avoid over-programming the cells in the memory matrix.

After double-checking some semantics as implied above, the manuscript can be published as is.

Author Response

Dear Reviewer 4

Thank you for delicately reviewing our manuscript. We have thoroughly revised the manuscript, considering all referee comments, and we have no disagreement with any comments. Furthermore, we marked up using the “Track Changes” function in MS Word to view any charges easily. We provide our response to all of your comments point-by-point. Please see the attachment.

Best regards,
Chanyang Park et al.

Round 2

Reviewer 1 Report

I thank the authors for their replies, and the replies satisfy me. I would like to recommend the publication of the current manuscript in the journal.